# Feasibility of a lung airway navigation system using fiber-Bragg shape sensing and artificial intelligence for early diagnosis of lung cancer

Lucian Gheorghe Gruionu[1]*, Anca Loredana Udriştoiu[2], Andreea Valentina Iacob[2], Cătălin Constantinescu[2], Răzvan Stan[1], Gabriel Gruionu[1,3]*

**1** Faculty of Mechanics, University of Craiova, Craiova, Romania, **2** Faculty of Automation, Computers and Electronics, University of Craiova, Craiova, Romania, **3** Division of Cardiology, Department of Medicine, Krannert Cardiovascular Research Center, Indiana University School of Medicine, Indianapolis, Indiana, United States of America

* ggruionu@iu.edu (GG); lgruionu@gmail.com (LGG)

**Data Availability Statement:** All relevant data are within the paper and its Supporting Information files.

## Abstract

Currently early diagnosis of malignant lesions at the periphery of lung parenchyma requires guidance of the biopsy needle catheter from the bronchoscope into the smaller peripheral airways via harmful X-ray radiation. Previously, we developed an image-guided system, iMTECH which uses electromagnetic tracking and although it increases the precision of biopsy collection and minimizes the use of harmful X-ray radiation during the interventional procedures, it only traces the tip of the biopsy catheter leaving the remaining catheter untraceable in real time and therefore increasing image registration error. To address this issue, we developed a shape sensing guidance system containing a fiber-Bragg grating (FBG) catheter and an artificial intelligence (AI) software, AIrShape to track and guide the entire biopsy instrument inside the lung airways, without radiation or electromagnetic navigation. We used a FBG fiber with one central and three peripheral cores positioned at 120° from each other, an array of 25 draw tower gratings with 1cm/3nm spacing, 2 mm grating length, Ormocer-T coating, and a total outer diameter of 0.2 mm. The FBG fiber was placed in the working channel of a custom made three-lumen catheter with a tip bending mechanism (FBG catheter). The AIrShape software determines the position of the FBG catheter by superimposing its position to the lung airway center lines using an AI algorithm. The feasibility of the FBG system was tested in an anatomically accurate lung airway model and validated visually and with the iMTECH platform. The results prove a viable shape-sensing hardware and software navigation solution for flexible medical instruments to reach the peripheral airways. During future studies, the feasibility of FBG catheter will be tested in preclinical animal models.

## Introduction

Early lung cancer diagnosis can improve survival rates by finding and sampling small malignant lesions in the lung parenchyma outside the lung airways. Currently, the large size of the bronchoscope prevents the operator to reach the peripheral pulmonary lesions [1].

**Funding:** "The research leading to these results has received funding from the Norwegian Financial Mechanism 2014-2021 under the project RO-NO2019-0138, 19/2020 "Improving Cancer Diagnostics in Flexible Endoscopy using Artificial Intelligence and Medical Robotics" IDEAR, Contract No. 19/2020."

**Competing interests:** The authors have declared that no competing interests exist.

Consequently, the biopsy needle is advanced outside the bronchoscope blindly or by X-ray guidance (fluoroscopy) exposing the patient and physician to harmful radiation [1].

In recent years, image-guided systems (IGS) were developed to help the physician visualize and target the surgical site during minimally invasive surgery procedures in neurosurgery, abdominal and thoracic surgery, intravascular intervention, and cardiac surgery [2–9]. IGS use pre-operative computed tomography (CT) or magnetic resonance imaging (MRI) image sequences to create a 3D map of the target anatomy, and electromagnetic or optical tracking systems to localize and track the position of surgical tools or therapeutic devices during the procedure [6, 9]. IGS has been used before by our team for tracking flexible medical instruments during the bronchoscopic and endoscopic procedures [7, 8, 10]. The computer software associated with the technology—iMTECH—performs the registration of the patient anatomical landmarks to the preoperative 3D map, real-time display of the position of the surgical tool in the 3D map, and assessment and corrections of differences between the preoperative imaging and intraoperative anatomical structures [7].

While the electromagnetic navigation can track the tip of the instrument, the surgeons cannot see the entire profile of the surgical tool inside the human body during procedures [11, 12]. They can only rely on the tool tip position, with inherent errors due to the image registration process, differences between pre- and live-procedure anatomy, and clinical and electromagnetic interferences. Also, the electromagnetic tracking systems cannot be used during MRI-guided procedures due to their metal content and interference with the magnetic field.

To address these limitations, a shape sensing system can improve the navigation during complex medical procedures. The Fiber Bragg Grating System (FBGS) is a shape sensing system with a small profile, light weight, no EM interference that can be embedded in many instruments [13]. The FBGS was used to build sensors array based on wavelength and space multiplexing [14]. FBG sensors have also been applied to colonoscopy for shape detection [13] and needle biopsy [15].

The commercially available FBGS are equipped with a flexible fiber and the software which determines the curvature or the relative 3D position of several points along the fiber. For a specific clinical application (e.g. lung cancer diagnosis) the user has to place the flexible fiber in a carrier instrument (e.g. a bronchoscope or a customized catheter) to navigate inside the body and develop additional software to determine the absolute position of the fiber with respect to anatomical landmarks (e.g. within lung airways). In the present study, we developed a new shape sensing system with a FBGS instrument, a custom-made carrier catheter and an artificial intelligence algorithm to navigate inside the lung airways. In addition, we developed a testing system to evaluate the navigation precision of the FBGS in an anatomically-accurate bench model of the lung airways. Its placement accuracy was evaluated visually and with an electromagnetic tracking software platform.

## Materials and methods

### Fiber-Bragg grating catheter system

A commercially available FBGS fiber (FBGS Technologies GmbH, Jena, Germany) was placed in the working channel of a custom-made three-lumen delivery catheter developed in our laboratory (Fig 1). The other two lumens contain a 5DOF Aurora electromagnetic (EM) tracking sensor to confirm the position of the tip of the FBGS fiber, and a guide wire with tip bending control to navigate specific airways during insertion.

The FBGS fiber has one central and three peripheral cores positioned at 120˚, with an array of 25 draw tower gratings (DTG) with 1cm/3nm spacing, 2 mm grating length, Ormocer-T coating, with a total outer diameter of 0.2 mm. The fiber is used with a MCF fan-out box and a

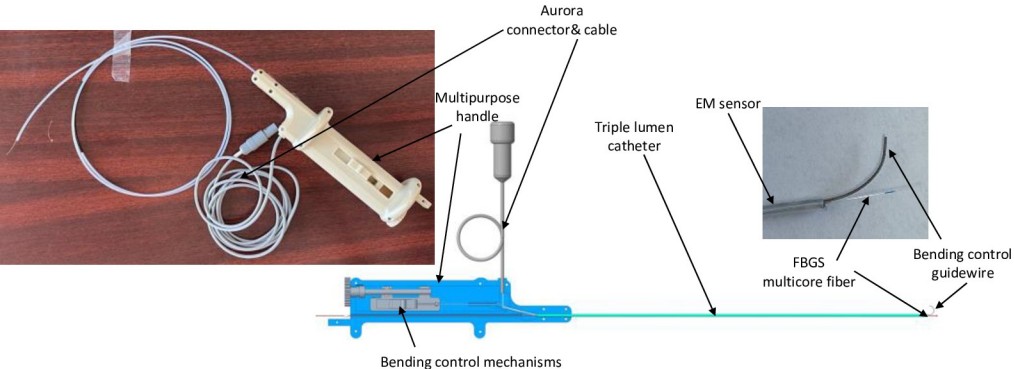

**Fig 1.** Schematic (a) and physical model (b) of the shape-sensing FBG system. 1- multipurpose handle; 2- three-lumen delivery catheter system; 3- Aurora EM sensor and wire; 4- bending control guidewire, 5 –bending control mechanism; 6 –the FBGS multicore shaper sensing fiber.

four-channel FBGS-Scan 904 interrogator that contains a broadband light source and a spectrometer. The FBGS ILLumiSense software was used to read out the spectral information from a laptop via an USB connection and calculate the peak wavelengths in real-time. The FBGS Shape Sensing software (version 1.3) was used to compute the curvature and 3D-shape of the array of gratings from the wavelength information obtained via TCP-streaming from the ILLumiSense software. The Shape Sensing software computes the curvature and shape data from the coordinates of the sensors/gratings with respect to the first proximal sensor using a method presented in [13] and provides the data in real time over a TCP port.

## Navigation software

A Python software AIrShape was developed to assist the doctor navigate in the lung airways using the shape-sensing catheter data from the FBGS fiber. The positioning data provided by the EM sensor inside the catheter are used to validate the results. The procedure steps are described in Fig 2.

A 3D computer model of the testing phantom was developed from a deidentified patient's CT stack by airways segmentation and reconstruction using the Mimics software (Materialise, Leuven, Belgium) and a CAD software, Creo Parametric (PTC, Boston, USA). The physical model was manufactured by 3D printing on 3D Systems MJP 2500 Plus printer (Rock Hill, USA), from the M2R-CL transparent polymer. On the CAD model, 46 median lines were

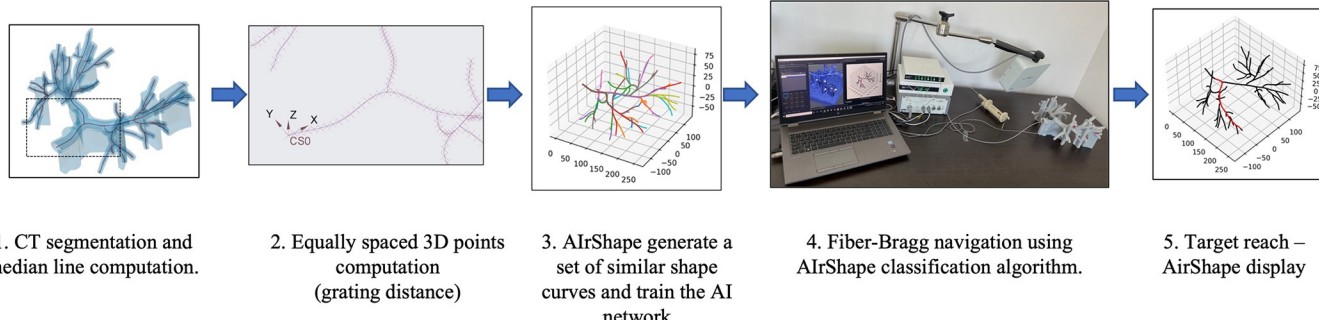

| 1. CT segmentation and median line computation. | 2. Equally spaced 3D points computation (grating distance) | 3. AIrShape generate a set of similar shape curves and train the AI network | 4. Fiber-Bragg navigation using AIrShape classification algorithm. | 5. Target reach – AirShape display |

**Fig 2. The steps of the airway navigation process using the FBGS shape-sensing fiber.** 1- Medial line computation; 2- Airway points (5 mm apart) placement along the median lines; 3- The software AIrShape generates a set of curves similar to the median lines to train the AI algorithm to identify the FBG catheter position; 4 –As the FBG catheter is advanced in the lung phantom, its shape is compared with the simulated curves and its position in a certain airway is identified; 5- AIrShape displays for median line position of the catheter for confirmation.

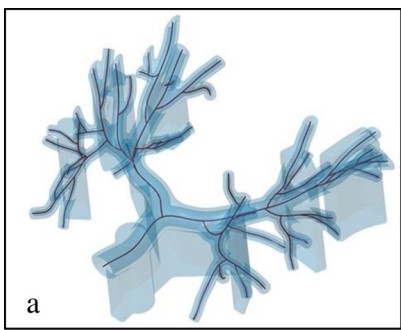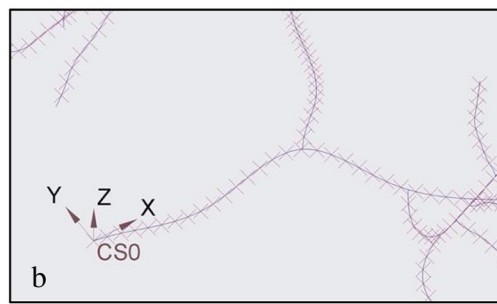

**Fig 3.** CAD model of the airway phantom (light blue) with median (dark red) lines (a), and equally spaced (5 mm) markers along the median lines in the rectangle caption (b). Similar 5mm markers are placed on the median lines of all lung airways.

identified (Fig 3A) starting from the trachea to the end of every peripheral branch. Equally spaced airway points 5mm apart were placed along the median lines using the Creo software, to be used for curves reconstruction and landmarks for navigation (Fig 3B).

The airway marker coordinates were uploaded in the AIrShape software, and the digital curves were reconstructed to be compared with the shape of the physical fiber using the artificial intelligence algorithm. If the digital curve of a median line is similar to the optical fiber shape, the segment is considered to represent the current position of the catheter in the airways and is overlaid on the lung airways (Fig 2A).

To test the system, we navigated 1–4 bifurcation orders to reach the peripheral branches (Fig 3A and numbered branches in Fig 4). For each peripheral branch navigation, the operator

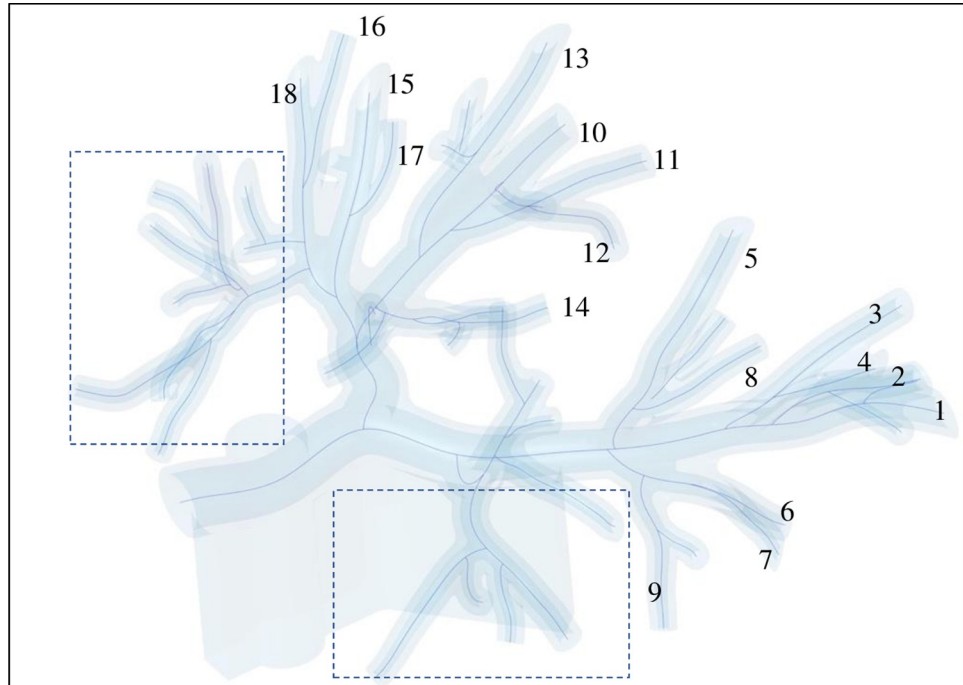

**Fig 4. Lung airways successfully navigated using the FBG catheter and the AIrShape software application (numbered 1–18).** Dashed line rectangles: upper lobe airways not navigated with this approach due to limitation of the bending radius of the FBG catheter.

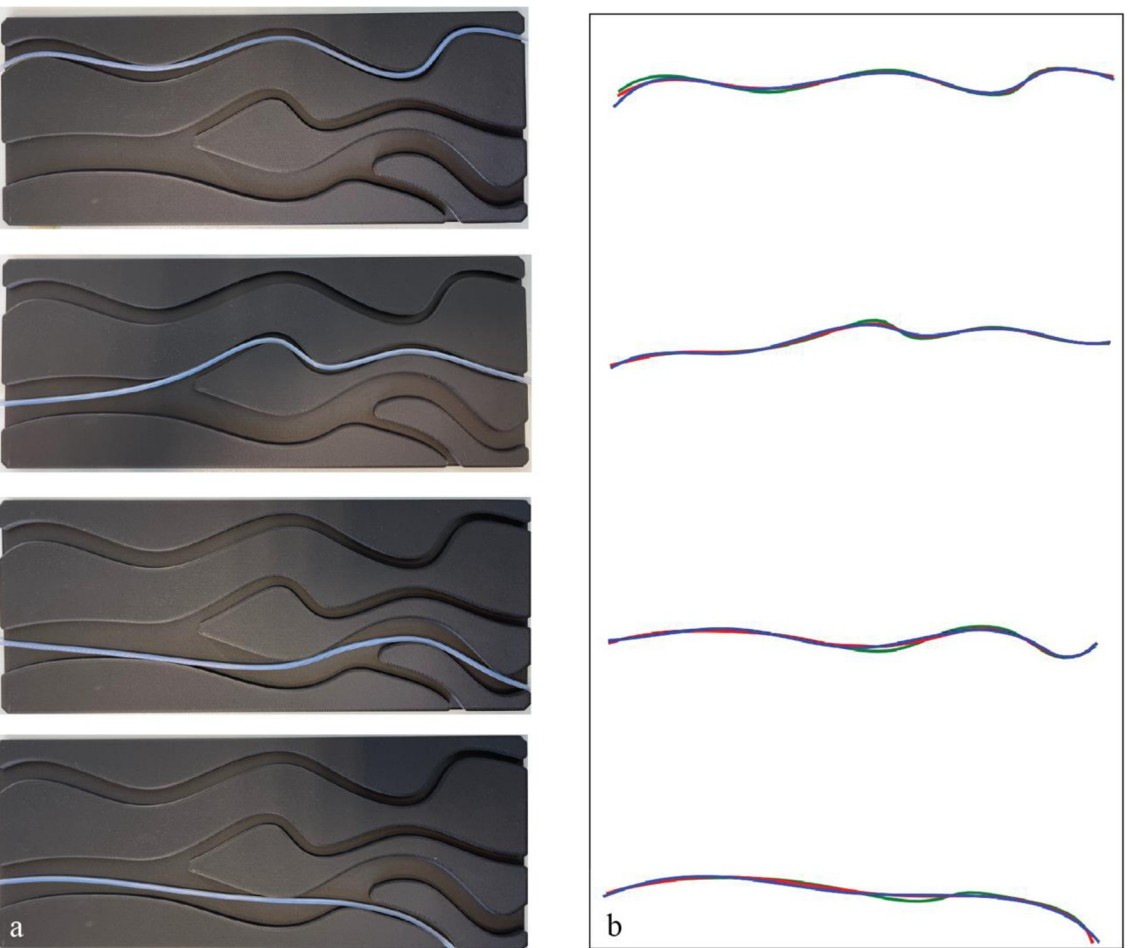

**Fig 5.** Test model to evaluate the FBG catheter's flexibility, placed in 4 channels (a) and computed/acquired curves which approximate the catheter position (b): median line of channels (green), computed shape of the catheter using AIrShape software (red) and FBGS shape sensing system (blue). The red and blue curves fit on top of each other and deviate slightly from the green curve (median line) due to the rigid catheter's tendency not to follow the median line.

repeated the fiber/catheter insertion 10 times. At each insertion, the coordinates of the fiber markers were generated automatically by the fiber software and run through the trained MVCNN software to identify the airway the fiber is located in.

## Catheter shape modeling

During navigation in the airways, the catheter does not follow the median line exactly due to its semi-rigid nature. To understand and model the catheter's deviation from the median line, a 2D simplified airway test model with four tortuous channels was customed designed and 3D printed (Fig 5A). The channels are 250–300 mm long with a width of 10 mm at entry to 4 mm at the end. The FBG catheter was placed in every channel (Fig 5A) and its shape acquired with the FBG system and Shape Sensing software.

The coordinates of markers along the median lines of channels and the corresponding channel width (*cw*) for every point were uploaded in the AIrShape software. First, the smoothing algorithm included in the AIrShape computes the curvature of the median lines [16].

For a parametrically defined curve,

$$\boldsymbol{r}(t) = x(t)\hat{\boldsymbol{i}} + y(t)\hat{\boldsymbol{j}} + z(t)\hat{\boldsymbol{k}}$$

the tangent vector is defined as:

$$\boldsymbol{T} = \frac{d\boldsymbol{r}}{dt}$$

and the unit tangent vector as:

$$\hat{\boldsymbol{T}} = \frac{velocity}{speed} = \frac{\boldsymbol{T}}{|\boldsymbol{T}|} = \frac{d\boldsymbol{r}/t}{|d\boldsymbol{r}/dt|}$$

where $\boldsymbol{v} = d\boldsymbol{r}/dt$ is the velocity of a point moving at a given time and the absolute value of the velocity vector is the speed vector of the curve $\boldsymbol{s}$, meaning:

$$|\frac{d\boldsymbol{r}}{dt}| = \frac{ds}{dt}$$

The curvature is defined as:

$$k = \frac{1}{|\boldsymbol{v}|}|\frac{d\hat{\boldsymbol{T}}}{dt}|$$

and the normal vector to a curve is the derivative of the tangent vector to a curve:

$$\boldsymbol{N} = \frac{d\hat{\boldsymbol{T}}}{ds}$$

The unit normal vector is

$$\hat{\boldsymbol{N}} = \frac{1}{k}\frac{d\hat{\boldsymbol{T}}}{ds}$$

To create multiple virtual AS curves similar to the catheter inserted in the test model channel (Fig 6), the AIrShape software computes the curvature of every point of the median line, and when $k > \alpha$, the points are translated along the normal axis $\boldsymbol{N}$ with a value

$$\boldsymbol{PT} = \gamma \, cw \, \hat{\boldsymbol{N}}$$

where $\gamma = random(0 \div (d_{max} - d_{min})/2)$ are random generated numbers, where $d_{max}$ and $d_{min}$ are the approximative maximum and minimum value of the airway's diameter/width. For the present testing model, $d_{max} = 16mm$, $d_{min} = 4mm$ and for the FBG catheter, $\alpha = 0.03$.

The newly computed points were interpolated and the resulting AS curve was smoothed out using the "splprep" and "splev" functions from the SciPy Python library respectively [17], and matched with the FBG fiber in the simplified airway test model (Fig 5A and 5B).

## Multi-view convolution neural network

The AIrShape software includes a multi-view convolutional neural network (MVCNN) to compare the 3D fiber shape with the median lines (Fig 6A). For every median line, the software creates multiple views by projecting it onto three orthogonal planes, resulting in three projection curves of 160x160 pixels for every 3D median line curve where 1 pixel = 1 mm$^2$ (Fig 6B).

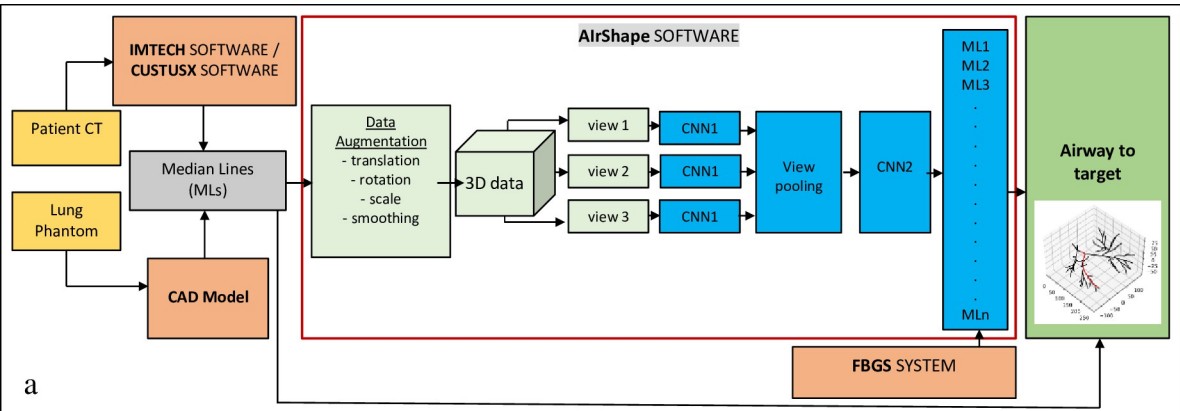

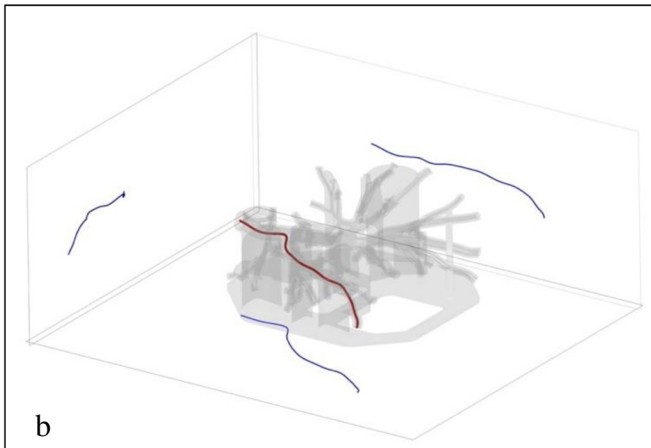

**Fig 6.** AIrShape MVCNN architecture and the complete navigation system data flow diagram from patient CT or lung phantom to median lines, AiRShape software and identification of airway trajectory to target (a), three orthogonal projections (views 1–3 above) computed for every median lines (b).

To create the training set for the MVCNN, for every median line, the AIrShape software computes 500 affined shape (AS) curves by smoothing according to the catheter flexibility using the catheter shape modeling algorithm from above.

We used an end-to-end 3x2D Multi-view Convolutional Neural Network (MVCNN) architecture for airways medial lines classification in lungs. For every one of the 500 AS curves, three orthogonal projections were computed resulting in a set of 1500 pictures for every median line of the 3D transparent model. The MVCNN contains a CNN for each of the three projections of each curve as inputs and applies convolution and max-pooling mathematical algorithms to extract their spatial features. Each CNN has eight layers: three convolutional layers, two pooling layers, one fully connected layer for each one of the three networks (Fig 7). At the end of the MVCNN simulation, the three networks are fully connected into one output layer (Fig 7). A rectified linear unit (ReLU) and SoftMax functions were used as activation functions in the convolution and fully connected layers, respectively. The parameters of the CNN network are listed in Table 1.

## Results and discussion

During the catheter navigation (Fig 2, step four of the shape sensing procedure), the catheter with the optical fiber is inserted in the phantom through the working channel of a

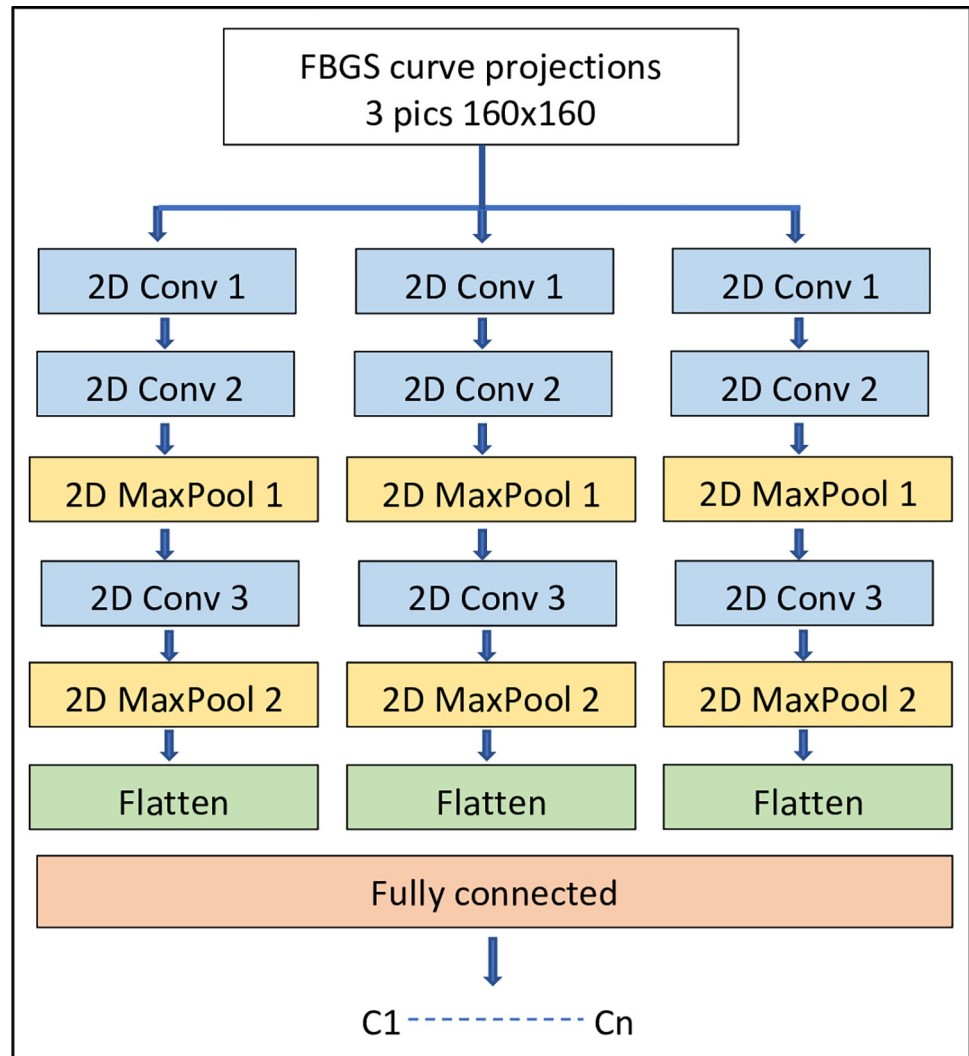

**Fig 7. The MVCNN architecture for identification of the median lines of lung airways.** Each median line is approximated with 500 AS curves and each AS curve is projected in three planes. For each projection, a CNN network is developed to identify its features. Individual CNN are fully connected into one output layer.

**Table 1. The parameters of the 2D CNN model for the three projections of each testing curve which approximates the position of the FBG catheter.**

| Layer name | Kernel dimensions (pixels, pixels) | Output dimensions (pixels, pixels, no. of kernels) |
|---|---|---|
| Conv2D 1 | (64,64) | (97,97,32) |
| Conv2D 2 | (12,12) | (86,86,64) |
| MaxPooling 2D 1 | (2,2) | (43,43,64) |
| Conv2D 3 | (3,3) | (41,41,128) |
| MaxPooling 2D 2 | (2,2) | (20,20,128) |
| FC 1 | - | 128 |
| FC 2 | - | 18 |

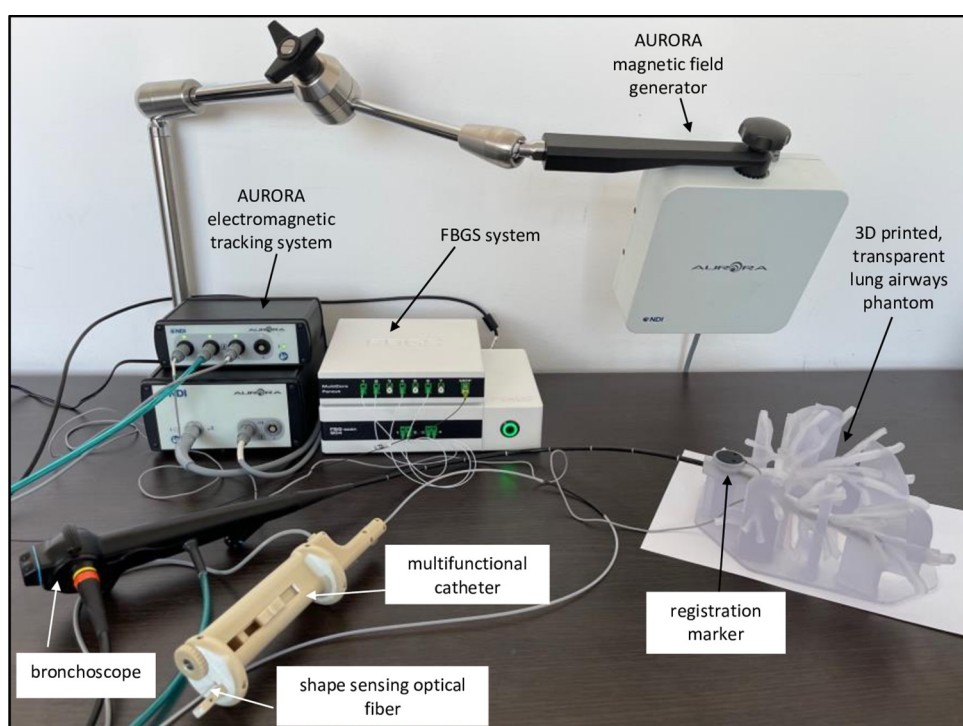

**Fig 8. AIrShape system setup for testing the shape sensing optical fiber in a custom-made delivery catheter.**

bronchoscope and extended into the lung airways to the periphery where the bronchoscope is too large to navigate (Fig 8). The position of the catheter inside the airway is displayed live using the AIrShape software (Fig 9A).

If the catheter is advanced in the wrong airway, it is retracted, rotated, bent and advanced in the correct branch. Its position is confirmed on the AIrShape screen (Fig 9A). The AI

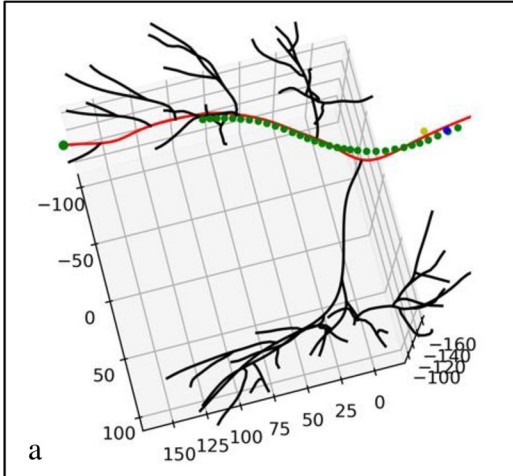
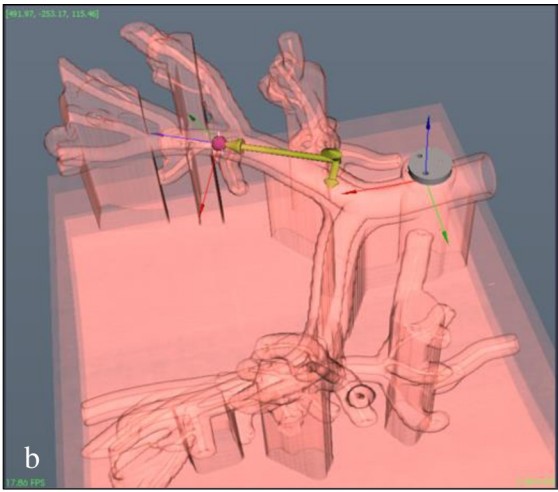

**Fig 9. Output of the AIrShape software.** a: software display of the shape-sensing catheter position (green dot line) in the correct lung airway (red line) and the remaining airways (black lines). b: the position of the FBG catheter's tip is confirmed with the iMTECH platform (red circle). The reference marker (grey circle), initial entry point (green circle) and the coordinate system (green arrows) are also shown.

**Table 2. The precision accuracy of identifying the correct lung airway by MVCNN during FBG navigation.** The results represent the mean and standard deviation of ten experiments.

| Airway Number | Mean Prediction Accuracy (%) | STDEV |
|---|---|---|
| 1 | 98.1 | 1.0 |
| 2 | 85.2 | 2.4 |
| 3 | 86.1 | 1.1 |
| 4 | 88.0 | 1.6 |
| 5 | 94.0 | 2.0 |
| 6 | 98.1 | 0.5 |
| 7 | 88.9 | 0.9 |
| 8 | 85.2 | 1.2 |
| 9 | 98.0 | 0.9 |
| 10 | 95.2 | 1.3 |
| 11 | 93.2 | 1.5 |
| 12 | 96.0 | 1.1 |
| 13 | 92.3 | 0.6 |
| 14 | 90.4 | 1.4 |
| 15 | 82.0 | 2.4 |
| 16 | 83.9 | 1.5 |
| 17 | 89.9 | 0.7 |
| 18 | 94.2 | 1.9 |
| Overall Mean | 91.0 | |
| Overall STDEV | 5.0 | |

algorithm computation is instantaneous, and the identified airway segment is highlighted in red over the lung airway tree (Fig 9A).

The image-guided (IM) software application previously developed by our group [7], iMTECH was used to validate the placement obtained with the FBGS system and the AIrShape software. The application uses the CT and electromagnetic tracking with AURORA system (Northern Digital Inc, Toronto, Canada) to identify the position of the medical instrument tip inside lung airways during the bronchoscopy procedure (Fig 9B). The position of the electromagnetic sensor placed on the tip of the delivery catheter is displayed on the 3D map to confirm the results from the AIrShape application (Fig 9B).

The result of the MVCNN model is the prediction accuracy percentage for each airway number (Table 2). The overall prediction accuracy average is 91% with a standard deviation of 5%.

The iMTECH software was used to confirm the FBG results. The position of the catheter's tip was displayed on the 3D map of the lung and the airway on screen compared to the airway identified on AIrShape software.

The AIrShape system includes a FBG shape-sensing catheter and an AI software to assist doctors during lung airway navigation with the purpose of locating peripheral lesions beyond the range where the bronchoscope can reach. As the doctor advances the FBG catheter into the airways, the model matches the trajectory of the FBG catheter to the 3D model of lung airways. Once the bronchoscope reaches the smallest diameter airway, the FBG catheter is extended outside the bronchoscope towards the target lesion which was identified on the lung CT. Once the target is reached, the biopsy instrument is advanced to the target and the biopsy is collected.

An EM sensor is not necessary for using the fiber-Bragg—AI system we developed. For navigation during bronchoscopy, the AIrShape and the FBG shape sensing technology can be

used as a standalone system to identify the airway the catheter is passing-through after bronchoscope extension, without electromagnetic tracking, video, or fluoroscopy assistance. In this case, the delivery catheter can be dual-lumen, one lumen for the bending tip guidewire and one for the FBG fiber. Once the catheter has reached the target, the FBG fiber or the guidewire can be removed, and the open lumen used for diagnostic or treatment instruments.

In the present study, the segmentation and median line computation of the phantom lung airways was performed in the CAD software. During clinical procedures, the median line identification can be performed within a dedicated software such as iMTECH or the open-source Fraxinus software platform [18].

Due to the bending radius of the FBG catheter being too wide for the sharp angles required to navigate the upper lobe branches, the FBG catheter could not reach all airways. Another limitation of the existing design is the bending radius of the electromagnetic sensor which further reduced the angle that can be navigated. While the FBG fiber is flexible enough to be advanced through any sharp turn in the upper lobe it is too flexible to be used by itself.

## Conclusions

The main advantage of using the AIrShape system in clinical applications is its precision of finding peripheral lung lesions when the biopsy instrument is extended outside of the bronchoscope. By monitoring the entire body of the catheter, the user makes sure the instrument is in the clinically relevant airway and is oriented in the right direction before the biopsy needle is advanced into the parenchymal tissue to collect the biopsy. The real-time shape visualization of the instrument reveals any catheter bending and avoids airway wall injury, pneumothorax, infection, and other clinical complications.

During the next phase of development, we plan to extend the AIrShape reach by using a more flexible delivery catheter and no electromagnetic sensor. Alternatively, the bending tip of the bronchoscope can be used to make the first sharp turn and then the delivery catheter advanced further to the periphery of the upper lobe. These new approaches will be tested in the lung airway model and preclinical animal studies followed by clinical testing.

## Supporting information

**S1 Data.**
(XLSX)

## Author Contributions

**Conceptualization:** Lucian Gheorghe Gruionu.

**Data curation:** Cătălin Constantinescu, Răzvan Stan.

**Formal analysis:** Cătălin Constantinescu, Răzvan Stan, Gabriel Gruionu.

**Funding acquisition:** Lucian Gheorghe Gruionu, Gabriel Gruionu.

**Investigation:** Răzvan Stan, Gabriel Gruionu.

**Methodology:** Lucian Gheorghe Gruionu, Anca Loredana Udriştoiu, Cătălin Constantinescu, Gabriel Gruionu.

**Project administration:** Lucian Gheorghe Gruionu, Andreea Valentina Iacob.

**Resources:** Andreea Valentina Iacob.

**Software:** Anca Loredana Udriştoiu.

**Supervision:** Lucian Gheorghe Gruionu.

**Validation:** Lucian Gheorghe Gruionu, Anca Loredana Udriştoiu, Cătălin Constantinescu, Răzvan Stan, Gabriel Gruionu.

**Visualization:** Andreea Valentina Iacob.

**Writing – original draft:** Lucian Gheorghe Gruionu, Gabriel Gruionu.

**Writing – review & editing:** Lucian Gheorghe Gruionu, Gabriel Gruionu.

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
