## [Decision Letter · Decision Letter 0]

2 Sep 2022

PONE-D-22-19712Feasibility of a Lung Airway Navigation System Using Fiber-Bragg Shape Sensing and Artificial Intelligence for Early Diagnosis of Lung CancerPLOS ONE

Dear Dr. Gruionu,

Thank you for submitting your manuscript to PLOS ONE. After careful consideration, we feel that it has merit but does not fully meet PLOS ONE’s publication criteria as it currently stands. Therefore, we invite you to submit a revised version of the manuscript that addresses the points raised during the review process.

We look forward to receiving your revised manuscript.

Kind regards,

Talib Al-Ameri, Ph.D

Academic Editor

PLOS ONE

Journal Requirements:

2. Please note that PLOS ONE has specific guidelines on code sharing for submissions in which author-generated code underpins the findings in the manuscript. In these cases, all author-generated code must be made available without restrictions upon publication of the work. Please review our guidelines at https://journals.plos.org/plosone/s/materials-and-software-sharing#loc-sharing-code and ensure that your code is shared in a way that follows best practice and facilitates reproducibility and reuse. New software must comply with the Open Source Definition.

3. Please include your tables as part of your main manuscript and remove the individual files. Please note that supplementary tables (should remain/ be uploaded) as separate "supporting information" files.

4. Please update your submission to use the PLOS LaTeX template. The template and more information on our requirements for LaTeX submissions can be found at http://journals.plos.org/plosone/s/latex.

"The research leading to these results has received funding from Norwegian Financial Mechanism 2014-2021 under the project RO-NO-2019-0138, 19/2020 “Improving Cancer Diagnostics in Flexible Endoscopy using Artificial Intelligence and Medical Robotics” IDEAR, Contract No. 19/2020."

"LGG, ALU, AVI, CC, RS, GG received funding from the the Norwegian Financial Mechanism 2014-2021 under the project RO-NO-2019-0138, 19/2020 “Improving Cancer Diagnostics in Flexible Endoscopy using Artificial Intelligence and Medical Robotics” IDEAR, Contract No. 19/2020.

https://uefiscdi.gov.ro\\

7. In your Data Availability statement, you have not specified where the minimal data set underlying the results described in your manuscript can be found. PLOS defines a study's minimal data set as the underlying data used to reach the conclusions drawn in the manuscript and any additional data required to replicate the reported study findings in their entirety. All PLOS journals require that the minimal data set be made fully available. For more information about our data policy, please see http://journals.plos.org/plosone/s/data-availability.

Reviewers' comments:

Reviewer's Responses to Questions

**Comments to the Author**

1. Is the manuscript technically sound, and do the data support the conclusions?

Reviewer #1: Yes

2. Has the statistical analysis been performed appropriately and rigorously? 

Reviewer #1: Yes

3. Have the authors made all data underlying the findings in their manuscript fully available?

Reviewer #1: Yes

4. Is the manuscript presented in an intelligible fashion and written in standard English?

Reviewer #1: Yes

5. Review Comments to the Author

Reviewer #1: Thanks for submitting this fabulous manuscript to plosone journal, please only rearrange the abstract in structural fashion, in result section the paragraph (We navigate order 1-4 order.........the fiber is located in. ) to method section

6. PLOS authors have the option to publish the peer review history of their article (what does this mean?). If published, this will include your full peer review and any attached files.

Reviewer #1: No

---

## [Author Response · Author response to Decision Letter 0]

10 Oct 2022

Thank you and the reviewer #1 for the careful consideration of our manuscript and informative comments. In response, please consider the following response to reviewer:

Reviewer #1: Thanks for submitting this fabulous manuscript to plosone journal, please only rearrange the abstract in structural fashion, in result section the paragraph (We navigate order 1-4 order.........the fiber is located in.) to method section.

A: 1. We rearranged the Abstract to correspond better to an Introduction/Methods/Results structure. A formal structure with separate paragraphs and titles for Introduction/Methods/Results is not allowed by the Journal according to the instructions at https://storage.googleapis.com/plos-published-prod/wjVg/PLOSOne_formatting_sample_main_body.pdf

2. We have moved the paragraph (We navigate order 1-4 order.........the fiber is located in. ) to the Methods section at lines 161-166 (yellow highlight) as instructed.

---

## [Decision Letter · Decision Letter 1]

8 Nov 2022

Feasibility of a Lung Airway Navigation System Using Fiber-Bragg Shape Sensing and Artificial Intelligence for Early Diagnosis of Lung Cancer

PONE-D-22-19712R1

Dear Dr. Gruionu,

We’re pleased to inform you that your manuscript has been judged scientifically suitable for publication and will be formally accepted for publication once it meets all outstanding technical requirements.

Kind regards,

Talib Al-Ameri, Ph.D

Academic Editor

PLOS ONE

Reviewers' comments:

Reviewer's Responses to Questions

**Comments to the Author**

1. If the authors have adequately addressed your comments raised in a previous round of review and you feel that this manuscript is now acceptable for publication, you may indicate that here to bypass the “Comments to the Author” section, enter your conflict of interest statement in the “Confidential to Editor” section, and submit your "Accept" recommendation.

Reviewer #1: All comments have been addressed

2. Is the manuscript technically sound, and do the data support the conclusions?

Reviewer #1: Yes

3. Has the statistical analysis been performed appropriately and rigorously? 

Reviewer #1: Yes

4. Have the authors made all data underlying the findings in their manuscript fully available?

Reviewer #1: Yes

5. Is the manuscript presented in an intelligible fashion and written in standard English?

Reviewer #1: Yes

6. Review Comments to the Author

Reviewer #1: Thanks for submitting this work to the journal, and thanks again for your efforts in taking the reviewer comments in your consideration, waiting for submission next work good luck.

7. PLOS authors have the option to publish the peer review history of their article (what does this mean?). If published, this will include your full peer review and any attached files.

Reviewer #1: No

---

## [Editor Report · Acceptance letter]

10 Nov 2022

PONE-D-22-19712R1 

Feasibility of a Lung Airway Navigation System Using Fiber-Bragg Shape Sensing and Artificial Intelligence for Early Diagnosis of Lung Cancer 

Dear Dr. Gruionu:

I'm pleased to inform you that your manuscript has been deemed suitable for publication in PLOS ONE. Congratulations! Your manuscript is now with our production department. 

Kind regards, 

on behalf of

Dr. Talib Al-Ameri 

Academic Editor

PLOS ONE